# Food safety practices of individuals before and after the emergence of COVID-19: A pre- and post-comparative analysis

Daniel Teshome Gebeyehu[1,2*], Leah East[1,3,4], Stuart Wark[5], Md Shahidul Islam[1]

1 School of Health, University of New England, Armidale, Australia, 2 School of Veterinary Medicine, Wollo University, Amhara Ethiopia, 3 Centre for Health Research, University of Southern Queensland, Toowoomba, Queensland, Australia, 4 School of Nursing and Midwifery, University of Southern Queensland, Toowoomba, Queensland, Australia, 5 School of Rural Medicine, University of New England, Armidale, Australia

* daniel.teshome@wu.edu.et

## Abstract

### Background

Recent literature highlights both beneficial and adverse effects of COVID-19 on individual food safety practices; however, the actual impact remains unverified, especially in low-resource countries. This study primarily aimed to investigate how COVID-19 has directly and indirectly influenced food safety practices among individuals in Ethiopia.

### Method

A retrospective survey was used to collect information related to the food safety practices of individuals, both before and after COVID-19. The survey was conducted in one metropolitan city, Addis Ababa, and three regional cities -Dessie, Kombolcha, and Debre Birhan - situated in the Amhara regional state of Ethiopia, between 16 April to 30 June 2023. The food buyers in the selected cities were randomly selected based on their place in queue in randomly selected food shops. As this study aimed to assess the impact of COVID-19 on individuals' food safety practices in both metropolitan and regional contexts, an equal number of participants were allocated and surveyed from both location types. IBM SPSS Version 28 was used for both data entry and statistical analyses. Following data entry, data cleaning and management were performed using SPSS Syntax commands to prepare the dataset for descriptive and ordinal logistic regression analyses.

### Results

Of the 396 sampled individuals, 51.5% were female and the remaining 48.5% were male. Participants' educational level had a statistically significant impact on overall

**Data availability statement:** All relevant data are within the paper and its Supporting Information files.

**Funding:** The author(s) received no specific funding for this work.

**Competing interests:** The authors have declared that no competing interests exist.

food safety practices both before (p = 0.001, AOR = 0.017) and after (p = 0.001, AOR = 0.002) the emergence of COVID-19. Similarly, the type of work for income generation influenced food safety practices both pre- (p = 0.15, AOR = 0.21) and post- (p = 0.21, AOR = 0.324) COVID-19. Participants' location was significantly associated with their food safety practices only before the emergence of COVID-19 (p = 0.006, AOR = 4.906). Additionally, dummy variables related to living arrangements/family size showed a statistically significant association, with p-values ranging from 0.001 to 0.002 and AOR between 27.578 and 168.937. During both pre- and post-COVID-19 periods, all the dummy variables of cleaning-related predictive variables had significant association with food safety practices - before the pandemic, p-values ranged from 0.001 to 0.023, with AOR between 0.00 and 0.059 and after the pandemic, p-values ranged from 0.001 to 0.017, with AOR between 0.00 and 15.596. Among cooking-related practices, cooking raw food before consumption was significantly associated with food safety practices, with p = 0.004 and AOR = 0.002 before the pandemic, and p = 0.031 and AOR = 0.08 after the pandemic.

## Conclusions

This study found that the emergence of COVID-19 had a positive impact on individuals' food safety practices, as they adhered more closely to food safety standards during the pandemic compared to the pre-pandemic period. Policymakers, food safety regulators, governmental and non-governmental organizations, as well as academic and research institutions, are encouraged to develop an integrated food safety sustainability policy. This policy should aim to maintain the advancements in food safety practices that resulted from the implementation of COVID-19 prevention and control measures.

## Introduction

COVID-19 was initially identified as a global public health threat and subsequently characterized as a pandemic on 30 January and 11 March 2020, respectively [1]. Regardless of geographic location and economic status, COVID-19 has and continues to severely affect individuals, countries, and global society in general [2–4]. COVID-19 resulted in a wide range of multi-sectorial crises, and problems such as food security and practices are likely to be exacerbated within lower- and middle-income countries, such as Ethiopia. As of 31 December 2023 (3 years after COVID-19 emergence), 773.82 million cases and approximately 7.1 million deaths globally have been reported to the World Health Organization (WHO), with over 7, 500 deaths and over half a million cases in Ethiopia [5]. However, as reported by Gebeyehu *et al*., [6], COVID-19 also had resulted in unexpected indirect positive health outcomes contrary to the otherwise devastating multi-dimensional impacts.

According to Djekic *et al*. [7], food safety practices improved during COVID-19 due to the strict application of COVID-19 infection prevention and control (IPC)

measures. This indicates that COVID-19 has an indirect and unexpected contribution to food safety. As described by previous studies [7,8], individuals' food safety awareness level and their hygienic practices were positively associated. In addition, a comparative study conducted in Indonesia and Bangladesh [9] corroborated that COVID-19 prevention measures, such as using face masks and gloves, improved individuals' food safety practices during the pandemic. Likewise, a study conducted in Kuwait [10] confirmed that COVID-19 positively impacted the hygienic and food safety practices of people. However, the emergence of COVID-19 diverted attention of food safety regulators to COVID-19 prevention and therefore regular food safety control systems were disrupted. As a result, food safety was compromised due to breaches and food safety regulation violations along the food supply chains during the pandemic [11]. As reported by Ma *et al*., [12], the COVID-19 pandemic negatively impacted the food safety sector due to disruption in food supply chains and food market restrictions.

## Conceptual model

Given the conflicting findings of previous research on the effects of COVID-19 on food safety practices - both positive [7,8] and negative [11,12] - the potential impacts of COVID-19 on individuals' food safety practices, along with its indirect effects on health, are conceptualized in Fig 1.

COVID-19 has multi-dimensional negative impacts on a variety of sectors worldwide and its impact on food safety remains unclear. The apparent contradiction between the positive [7,8] and the negative food safety effects of COVID-19 [11,12] indicates that further studies are required to determine the real impact (negative, positive, or both) of COVID-19 on food safety. The previous studies did not assess pre-COVID-19 food safety status and only focused on the post-pandemic food safety practices. To evaluate the impact of COVID-19 on food safety, food safety practices both before COVID-19 and after its emergence are needed. To facilitate effective future planning, prioritizing, and allocating food safety resources in lower- and middle-income countries, it is imperative to assess the impact of COVID-19 on food safety. As such, this study focused on the direct and indirect impact of COVID-19 on food safety practices in metropolitan and regional contexts of Ethiopia, and with consideration both pre- and post-COVID-19.

## Methods

### Study area

The study was conducted in Addis Ababa (a metropolitan city) and three regional cities of Amhara regional state (Dessie, Kombolcha, and Debre Birhan) in Ethiopia. Addis Ababa is the most populous city in Ethiopia with an estimated population number of 3,774,000, while Dessie, Kombolcha, and Debre Birhan have respective population numbers of 257,126; 125,654; and 139,724 [13]. According to the Ethiopian Ministry of Health, COVID-19 was more prevalent in Addis Ababa than in other regional cities.

To compare the food safety impacts of COVID-19 between the metropolitan city (Addis Ababa) and regional cities, three sites (Dessie, Kombolcha, and Debre Birhan) were purposively selected from Amhara region of Ethiopia. As a consequence of research budget limitations, and to minimise any impacts associated with language barriers, the three regional cities were chosen based on their proximity to the metropolitan city and the type of language (Amharic) the participants spoke.

### Study population

The study's participants examining COVID-19's impact on food safety were individuals randomly selected from open-air food markets in Addis Ababa and the three cities in Amhara regional state. In the context of this study, an open-air market was defined as a public marketplace where food items and other goods are sold in different mini shops and open-air spaces. All markets in both metropolitan and regional cities were listed, individually numbered by the lead author, and

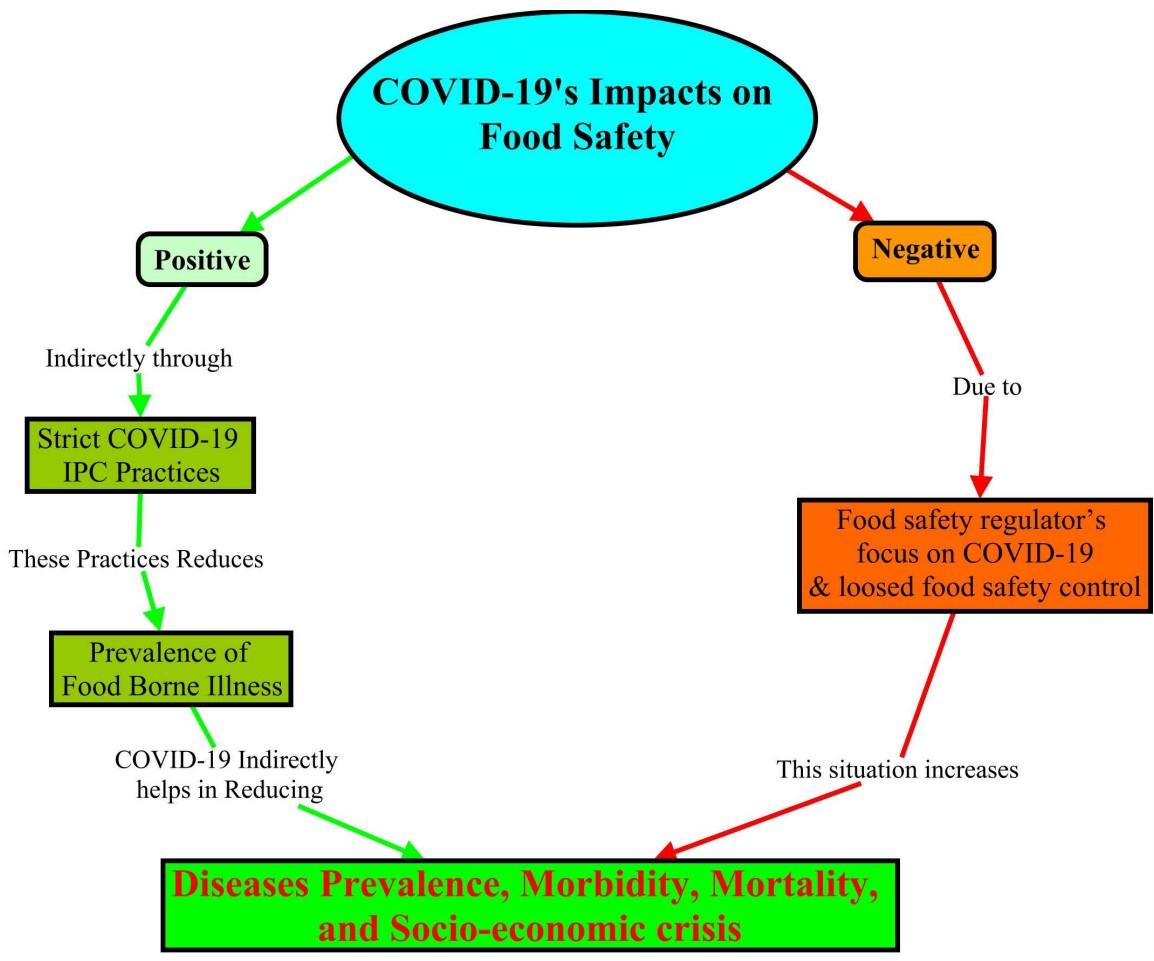

**Fig 1. Concept map of COVID-19's impact on food safety and its indirect impact on health based on the findings of previous studies.**

three open-air markets from the metropolitan city and one from each regional city were then randomly selected using the lottery method. Each chosen market had distinct sections dedicated to different food items, including vegetables and fruits, cereals, live animals and animal products, spices, and packed or processed foods. Within each section there were multiple shops, which were again numbered by the lead author, with one shop from each section being randomly selected using the lottery method. As a result, 6 shops from the 6 food item sections of 1 open-air market and 36 shops from the 6 open-air markets were selected. The lead data collector asked permission from the selected shop owner to undertake this research prior to commencement. Two data collectors collaborated during the data collection process, with one administering the survey to participants while the other managed the order of potential participants in the queue formed by food buyers at the cashier or shop clerk. Every 10th food buyer who consented to participate and was over 18 years old was selected for the survey. Since participants were chosen from the existing queue leading to the billing area, individuals outside this queue were not allowed to take part. Given the large number of people in an open-air market, creating a complete sampling frame was impractical. Following the recommendation of Mooney and Garber [14], the queue itself was used as the sampling frame in such large population size. An equal number of individuals were selected from each food item section. Additionally, to compare the impact of COVID-19 in regional and metropolitan settings, the number of participants from the metropolitan city (Addis Ababa) and regional cities was proportionally allocated.

## Study design, sample size, and sampling procedure

Since data on participants' prior knowledge of food safety practices needed to be gathered, a retrospective cross-sectional study design was employed. The survey was conducted from 16 April to 30 June 2023. Since this study utilized a retrospective cross-sectional design, the Strengthening Reporting of Observational Studies in Epidemiology (STROBE) reporting guideline was recommended [15] and adhered to in the preparation of this manuscript.

Participants were asked to recall their food safety practices and food security experiences from one year before and after the first reported case of COVID-19 in Ethiopia (13 March 2020). Each survey took 20–30 minutes to complete.

The sample size was calculated prior to commencement using Fisher's formula, as shown below.

$$n = \frac{Z^2 PQ}{d^2} = \frac{(1.96)^2 (0.5)(0.5)}{(0.05)^2} = 384$$

Where:
n = is the minimum sample size for a statistically significant survey
Z = is normal deviant at the portion of 95% confidence interval = 1.96
P = is status of individuals' food safety practices before and after COVID-19. Since there was no previous study on this topic in the study area, 50% prevalence rate was used.
Q = 1-P
d = is margin of error acceptable or measure of precision = 0.05
In addition to the above equation, the authors used the Australian Bureau of Statistics Sample Size Calculator [16] which recommended 385 participants. Since the difference between the two methods was negligible, the authors determined that the minimum survey sample size for this study would be the slightly higher figure of 385. Since data collection was conducted through face-to-face interviews by trained data collectors, rather than being self-administered, there were ultimately no non-respondents in the survey. If a participant changed their mind after initially agreeing to take part, the data collectors respected their decision and proceeded to the next eligible participant.

## Data collection, validation and management

**Study variables.** Of the 22 total independent variables, six were categorized under the *cleaning* food safety step, while the remaining three food safety steps - *separating*, *cooking*, and *chilling* - each included three independent variables. The remaining seven independent variables were classified under *demographic variables*.

The *cleaning* food safety step included six independent variables: washing hands before, during, and after food preparation; before and after eating; after touching raw food, shaking hands, or touching one's face; after using the toilet; using soap when washing hands, utensils, and food preparation surfaces; and using sanitizer when feeling contaminated.

The *separating* food safety step comprised three explanatory variables: keeping raw food separate from ready-to-eat food; storing seafood and meat in sealed containers with their juices contained in the refrigerator; and using separate cutting boards for raw and cooked foods.

The *cooking* food safety step was evaluated using three questions: cooking raw food before consumption; using appropriate cooking temperature guidelines for different food types; and thoroughly reheating stored food before consumption.

The *chilling* food safety step included three independent variables: storing cooked and uncooked food at the recommended refrigerator temperature; sealing food items separately before refrigeration; and promptly chilling perishable food.

Finally, *demographic variables* - including participants' residential location, age, gender, education level, marital status, household size, and employment status - were also considered as independent variables.

A *dependent* variable with three categorical responses - *sufficient, moderate, or optimum* - was derived from participants' frequency of compliance (*never*, *sometimes*, or *always*, respectively) with the handwashing practice questions in

the cleaning food safety step. Since handwashing has been recognized as the most critical food safety practice for food handlers [17,18], participants' handwashing practices at various stages of food preparation and consumption were pooled and used as an independent/outcome variable. The effect of each of the 22 independent variables on overall food safety practices was then analyzed and interpreted accordingly.

**Operational definition of terms and measurement.** According to the Australia and New Zealand Food Safety Standards [18], cleaning involves the physical removal of contaminants using water and/or detergents but does not eliminate pathogens. Meanwhile, the Centre for Diseases Control and Prevention [19] defines the separating step of food safety as keeping raw and cooked food, as well as food preparation equipment and surfaces, separate to prevent cross-contamination. In contrast to cleaning, the cooking step in food safety not only removes pathogens but also uses heat to destroy them, ensuring the food is safe to eat [19]. Finally, the chilling food safety step is the process of lowering food temperature to a safe level to inhibit the growth of harmful pathogens [19].

The study evaluated food buyers' food safety practices before and after COVID-19 to assess the pandemic's impact. These practices were measured based on the four food safety steps recommended by the CDC [19]: *cleaning* (frequent hand and surface washing and disinfecting); *separating* (preventing cross-contamination); *cooking* (ensuring proper food temperatures); and, *chilling* (prompt refrigeration) (S1 File).

Following the approach of previous research [20–23], individuals' food safety practices before and during COVID-19 were assessed using a three-level Likert scale: *never, sometimes, and always*. If a participant answered *never* to a question, their food safety practice for that item was categorized as *insufficient*. A *sometimes* response was classified as *moderate*, while an *always* response was rated as *optimum*. The same assessment method was used for both pre- and post-COVID-19 food safety evaluations. Previous studies have categorized food safety practices as *"poor, moderate, and good"* [24] or *"low, medium, and high"* [25]. However, this study adopted the neutral terms *insufficient, moderate, and optimum* for food safety practice categorization to avoid judgemental terminologies. Ordinal logistic regression was employed to examine the effect of each food safety practice item within the four food safety steps, as well as the demographic variables of the participants.

A comparison of pre- and post-COVID-19 responses was conducted to determine how the pandemic influenced food safety practices. If a participant had previously answered *never* or *sometimes* but responded *always* after COVID-19, their food safety practices were considered improved due to the pandemic's direct or indirect impact. Conversely, if a participant's response changed from *always* or *sometimes* before COVID-19 to *never* afterward, the pandemic was considered to have negatively affected their food safety practices.

**Pilot testing.** Before data collection, a pilot study was conducted with 70 participants from an open-air market (different from the main study locations). Sample size recommendations from various researchers were considered: one [26] suggested 24 participants, while others [27] and [28] recommended 55 + and 70, respectively. Based on the latter two recommendations, 70 participants were selected for the pilot study.

The internal consistency of the survey was evaluated using *Cronbach's alpha coefficient*, where values ≥0.6 were deemed reliable. The Cronbach's alpha values were 0.9 for pre-COVID-19 and 0.7 for post-COVID-19 food safety assessments, indicating good reliability. Two survey questions with Cronbach's alpha values below 0.6 were removed based on the pilot study findings.

## Study biases and mitigation strategies

One expected bias was selection bias, as the study focused on food buyers present at open-air markets during data collection. This method may have excluded individuals with significant food safety practices relevant to the study who were not at the markets at that time. To mitigate this selection-related bias and include a diverse range of individuals with various food safety practices before and after the emergence of COVID-19, a random sampling technique was employed.

 

Another potential bias was recall bias, stemming from participants' inability to accurately remember their food safety practices from one year before and after the onset of COVID-19. To limit the effect of this bias on the finding of this study, participants with responses marked as "I don't remember" were excluded during the data cleaning process.

The third anticipated bias was social desirability bias, where participants might overstate, underreport, or selectively offer responses perceived as socially acceptable. To reduce this risk, participants were informed beforehand that there would be no penalties for negative responses or rewards for positive ones. Additionally, the survey was conducted in private settings to prevent participants from being influenced by others around them. These private spaces varied by location - some interviews took place in enclosed rooms within stores, while others occurred in makeshift rooms created using curtain partitions at market sites. This setup ensured participants felt comfortable sharing honest answers.

Another possible bias was related to translation and transcription, which was minimized by selecting only Amharic-speaking participants, as this language was well understood by the data collectors.

While steps were taken to mitigate these biases, they could not be eliminated entirely. The limitations section of the research duly acknowledged their potential impact on the study's outcomes.

## Ethics approval and consent to participate

Research ethics approval was gained from the University of New England Human Research Ethics Committee (HREC), with approval number: HE22–173. After the project received ethical approval from the University of New England Human Research Ethics Committee (UNE-HREC), we requested additional ethics approval from the first author's home institution in Ethiopia (Wollo University). The "Wollo University Institutional Research Ethics Review Committee" approved the ethics clearance granted by UNE-HREC. For the purposes of confidentiality, any personal identifiers such as the names of the participants were not recorded. The objective of the data collection was provided to the participants, and they signed a written consent form before the commencement of the survey.

## Data analysis

Data entry and statistical analyses were carried out using IBM SPSS Version 28. Following data entry, data cleaning and management procedures were performed using SPSS Syntax commands in preparation for statistical analysis. These commands are particularly effective for identifying and correcting errors or inconsistencies related to data formatting and values, including missing data, skips, misinterpretations, duplicates, and typographical errors. Descriptive statistical analysis using percentages, numbers, tables, and graphs was applied. The respondents' food safety practices before and during COVID-19 were compared and analyzed using both descriptive and inferential statistics. Given that the dependent variable was ordinal, the authors employed ordinal logistic regression to assess the impact of each food safety question on the overall food safety practice (dependent variable).

Before choosing the statistical analysis, four assumptions (ordinality level of the dependent variable; presence of ordinal, continuous, categorical, or dichotomous independent variables; no/absence of multicollinearity; and presence of proportional odds ratio) for ordinal regression were tested. The first two assumptions were tested using a detailed understanding of the nature of the data and the remaining two were tested using SPSS software. The Polytomous Universal Model (PLUM) procedure was used in ordinal logistic regression. The variables with p-value < 0.05 were taken as variables that demonstrated important association. To mitigate confounding biases from multiple predictive variables and assess the relationship between independent and dependent variables, the adjusted odds ratio (AOR) with a 95% confidence interval was utilized. In addition, the Wald Chi-Squared Test (Wald/$X^2$) was employed to identify statistically significant explanatory (independent) variables. The explanatory variables with Wald's value different from nil (zero) were considered as significant to the model (food safety practice analysis). Conversely, if Wald's is equal to zero, the null hypothesis is true, and the variable is rejected from the model. For independent variables with more than two groups, the

omnibus test was conducted to analyze the statistical significance of each group, and each group under each predictive variable was treated as a dummy variable. Dummy variables, also known as indicator variables, are used in both descriptive and inferential analysis. They are used to test the presence or absence of a categorical effect on the variable to be tested (the dependent variable).

## Results

Of the total 396 individuals surveyed for their food safety practices before and after the emergence of COVID-19, half (198) of them were from a metropolitan city (Addis Ababa) and the remaining half (198) were from regional cities (Dessie, Kombolcha, and Debre Birhan).

### Effect of demographic factors on individuals' food safety practices before and after the emergence of COVID-19

Before the emergence of COVID-19, the individuals' location (Addis Ababa, OR = 4.906 and Kombolcha, OR = 5.044), their educational status (no formal education, OR = 0.017), and all dummy variables of the variable's living arrangement and work for income generation had statistically significant association with their food safety practices (Table 1). Their location and living arrangement were not statistically significant in the post-COVID-19 food safety assessment. All dummy variables (no formal education, primary school, and secondary school) of the educational level variable were statistically significant, with AOR of 0.002, 0.066, and 0.046, respectively, with reference to the dummy variable 'higher education' during post-pandemic emergence, but not before emergence.

### Effect of food safety items on the individuals' food safety practice both before and after COVID-19's emergence

All the cleaning-related food safety assessment questions (cleaning-related independent variables) had statistically significant associations with individuals' pre- and post-food safety practices (Table 2). Among the cooking-related variables, the raw food cooking practice of individuals both before (OR = 0.02) and after (OR = 0.08) the emergence of COVID-19 was significantly associated with their pooled food safety practices while the practice of individuals on 'thoroughly heating ready-to-eat foods before consumption' had a significant association (OR = 0.005) with the cumulative food safety practices of individuals before COVID-19 only (Table 2).

Food storage practice in the recommended temperature post-COVID-19 was significantly associated with their food safety practices with the AOR of 0.052 (never practice) and 0.245 (sometimes practice). The optimum food safety practices of individuals before the emergence of COVID-19 (21.5%) increased to 43.7%, during COVID-19 (Fig 2) which represents a 22.2% increase. Similarly, individuals' moderate and insufficient food safety practices reduced by 12.9% and 9.4% respectively during the COVID-19 pandemic (Fig 2).

## Discussion

The purpose of this study was to assess the impact of COVID-19 on the food safety practices of individuals in a lower- and middle-income country of Ethiopia using a pre-and post-comparative analysis. For this study, a total of 396 individuals were surveyed for both pre- and post-COVID-19 food safety practice assessments. As indicated in Table 2, the high number of 'never' practice responses of individuals before the emergence of COVID-19 reduced during the post-pandemic assessment. Inversely, the dwindling number of 'always' practiced responses before the pandemic has increased during COVID-19. This indicated that food safety was more widely practiced following the emergence of COVID-19. This finding is corroborated by previous findings [29–31] that individuals' food safety practices improved post-COVID-19. This is believed to be primarily due to heightened public health awareness, strict government enforcement of disease prevention measures, and widespread fears and concerns regarding the contagiousness and fatality of COVID-19 [32,33].

**Table 1. Frequency and percentage of demographic variables and their associations with the pooled pre and post COVID-19 emergence food safety practice of individuals using ordinal logistic regression.**

| Variables | Categories | Percentages (n = 396) | Pre-COVID-19 emergence | | Post-COVID-19 emergence | |
|---|---|---|---|---|---|---|
| | | | Wald (X²) | p value (AOR) | Wald (X²) | p value (AOR) |
| Location | Addis Ababa | 50 | 7.524 | 0.006(4.906) | 0.686 | 0.408(1.357) |
| | Dessie | 16.7 | 1.837 | 0.175(2.607) | 0.542 | 0.462(0.714) |
| | Kombolcha | 16.7 | 5.669 | 0.017(5.044) | 0.075 | 0.785(0.884) |
| | Debre Birhan | 16.7 | – | – | – | – |
| Age | 18-30 | 15.4 | 0.033 | 0.857(0.894) | 3.041 | 0.081(0.456) |
| | 31-40 | 26 | 0.300 | 0.584(1.349) | 1.041 | 0.308(0.673) |
| | 41-50 | 37.1 | 0.042 | 0.838(1.11) | 2.515 | 0.113(0.583) |
| | >50 | 21.5 | – | – | – | – |
| Gender | Male | 48.5 | 0.805 | 0.370(0.681) | 1.291 | 0.256(0.733) |
| | Female | 51.5 | – | – | – | – |
| Level of education | No formal education | 14.1 | 20.805 | 0.001(0.017) | 1.291 | 0.001(0.002) |
| | Primary school | 22.5 | 1.817 | 0.178(0.45) | 1.291 | 0.001(0.066) |
| | Secondary school | 32.3 | 2.190 | 0.139(0.482) | 1.291 | 0.001(0.046) |
| | Higher Education | 31.1 | – | – | – | – |
| Marital status | Unmarried | 29.5 | 2.752 | 0.097(3.768) | 1.000 | 0.317(1.638) |
| | Married | 51.3 | 2.050 | 0.152(0.405) | 0.03 | 0.862(0.943) |
| | Divorced | 19.2 | – | – | – | – |
| Living arrangement | No dependent | 33.6 | 21.392 | 0.001(168.937) | 3.579 | 0.059(2.496) |
| | Couple | 14.9 | 10.732 | 0.001(27.578) | 0.00 | 0.989(1.007) |
| | Have one child | 7.8 | 9.312 | 0.002(68.224) | 0.133 | 0.716(0.834) |
| | Have 2–4 children | 10.1 | 18.711 | 0.001(56.927) | 0.125 | 0.724(0.851) |
| | Have 5 + children | 33.6 | – | – | – | – |
| Work for income generation | Full time government employee | 15.7 | 5.870 | 0.015(0.21) | 5.344 | 0.021(0.324) |
| | Non-governmental organization employee | 16.2 | 15.802 | 0.001(105.639) | – | – |
| | Casual labourer | 15.4 | 8.340 | 0.004(0.069) | 16.955 | 0.001(0.154) |
| | Daily labourer | 38.1 | 20.549 | 0.001(0.01) | 3.845 | 0.05(0.431) |
| | Private business owner | 14.6 | – | – | – | – |

The dummy variables indicated by hyphen (-) is a reference category in the omnibus statistical significance test

### Effects of participants' demographic variables on their food safety practices

Contrary to the pre-COVID-19 food safety practice assessment, the location of participants was not associated with their food safety practices post-COVID-19 (Table 1). The statistical association between the location of individuals and their food safety practices before the emergence of COVID-19 might be due to the food safety awareness differences in various study areas [33]. However, after the onset of the pandemic, COVID-19 infection prevention measures were promoted and enforced regardless of individuals' locations. Consequently, the infection prevention practices adopted by most people during COVID-19 inadvertently enhanced their food safety habits. This contrasts with a study conducted in Thailand [34], which found that individuals' locations influenced their food safety practices after the emergence of COVID-19. This discrepancy may be attributed to differences in participants' levels of awareness, socio-economic status, access to information, cultural practices, and geographic locations.

**Table 2.  Frequency of individuals' food safety practices and the effect of independent variables (questions) on the pooled food safety status of individuals using ordinal logistic regression.**

| Questions | Frequency of practice | Pre-COVID-19 emergence | | | Post-COVID-19 emergence | | |
|---|---|---|---|---|---|---|---|
| | | Percentages (n = 396) | Wald (X²) | p-value (AOR) | Percentages (n = 396) | Wald (X²) | p-value (AOR) |
| **Cleaning related questions** | | | | | | | |
| Washing hands before, during and after food preparation. | Never | 28 | 9.072 | 0.003 (0.003) | 13.4 | 21.682 | 0.001 (0.00) |
| | Sometimes | 47.7 | 5.436 | 0.020 (0.032) | 32.3 | 18.146 | 0.001 (0.077) |
| | Always | 24.2 | – | – | 54.3 | – | – |
| Washing hands before and after eating. | Never | 24.5 | 15.367 | 0.001 (0.00) | 10.1 | 5.701 | 0.017 (0.004) |
| | Sometimes | 52.3 | 10.811 | 0.001 (0.006) | 24.7 | 2.889 | 0.089 (0.302) |
| | Always | 23.2 | – | – | 65.2 | – | – |
| Washing hand after touching uncooked food, shaking, and touching body parts. | Never | 26.5 | 11.618 | 0.001 (0.001) | 14.9 | 3.249 | 0.071 (0.114) |
| | Sometimes | 53.8 | 5.200 | 0.023 (0.059) | 30.8 | 5.670 | 0.017 (0.279) |
| | Always | 19.7 | – | – | 54.3 | – | – |
| Washing hands after toilet. | Never | 28.3 | 11.473 | 0.001 (0.001) | 15.2 | 13.261 | 0.001 (0.008) |
| | Sometimes | 50 | 9.265 | 0.002 (0.012) | 33.6 | 22.508 | 0.001 (0.066) |
| | Always | 21.7 | – | – | 51.3 | – | – |
| Using soap during hand, utensil and food preparation surfaces washing. | Never | 39.4 | 14.202 | 0.001 (0.00) | 14.1 | 5.757 | 0.016 (15.596) |
| | Sometimes | 41.2 | 10.779 | 0.001 (0.004) | 35.1 | 2.074 | 0.15 (0.436) |
| | Always | 19.4 | – | – | 50.8 | – | – |
| Using sanitizer when feeling contaminated. | Never | 56.6 | 10.277 | 0.001 (0.006) | 19.2 | 0.012 | 0.912 (1.102) |
| | Sometimes | 29.8 | 7.333 | 0.007 (0.009) | 33.1 | 6.834 | 0.009 (0.225) |
| | Always | 13.6 | – | – | 47.7 | – | – |
| **Separating related questions** | | | | | | | |
| Separate uncooked food from ready to eat food. | Never | 129.5 | 0.061 | 0.805 (0.666) | 15.2 | 0.118 | 0.732 (1.512) |
| | Sometimes | 43.4 | 0.498 | 0.480 (2.231) | 41.7 | 0.520 | 0.471 (0.589) |
| | Always | 27 | – | – | 43.2 | – | – |
| Storing sea food, and meat in sealed container together with their juices in the refrigerator. | Never | 32.1 | 0.254 | 0.614 (0.388) | 17.4 | 0.035 | 0.852 (0.809) |
| | Sometimes | 49.5 | 0.798 | 0.372 (3.653) | 46.5 | 0.043 | 0.837 (0.861) |
| | Always | 18.4 | – | – | 36.1 | – | – |
| Using separate cutting boards for uncooked and ready to eat foods. | Never | 61.9 | 0.090 | 0.764 (.537) | 37.4 | 2.071 | 0.15 (0.212) |
| | Sometimes | 32.6 | 0.214 | 0.643 (0.406) | 32.1 | 3.142 | 0.076 (0.166) |
| | Always | 5.6 | – | – | 30.6 | – | – |
| **Cooking related questions** | | | | | | | |
| Cooking raw food before consumption. | Never | 24.5 | 8.224 | 0.004 (0.002) | 13.9 | 4.628 | 0.031 (0.08) |
| | Sometimes | 50 | 10.983 | 0.001 (0.002) | 34.8 | 0.061 | 0.806 (0.868) |
| | Always | 25.5 | – | – | 51.3 | – | – |
| Using appropriate cooking temperature guide for different food types. | Never | 64.4 | 0.377 | 0.539 (3.702) | 35.9 | 2.536 | 0.111 (0.205) |
| | Sometimes | 28.8 | 0.002 | 0.963 (1.1) | 32.8 | 0.537 | 0.464 (0.496) |
| | Always | 6.8 | – | – | 31.3 | – | – |
| Thoroughly heating prepared and stored foods before consumption. | Never | 23.5 | 6.168 | 0.013 (0.005) | 10.9 | 3.679 | 0.055 (0.034) |
| | Sometimes | 49.7 | 0.007 | 0.933 (0.894) | 37.1 | 0.01 | 0.921 (1.060) |
| | Always | 26.8 | – | – | 52 | – | – |
| **Chilling related questions** | | | | | | | |
| Storing cooked and uncooked food in a recommended refrigerator temperature | Never | 16.4 | 0.001 | 0.977 (1.048) | 15.9 | 5.961 | 0.015 (0.052) |
| | Sometimes | 57.3 | 0.212 | 0.645 (1.749) | 41.7 | 4.665 | 0.031 (0.245) |
| | Always | 26.3 | – | – | 42.4 | – | – |

*(Continued)*

| Questions | Frequency of practice | Pre-COVID-19 emergence | | | Post-COVID-19 emergence | | |
|---|---|---|---|---|---|---|---|
| | | Percentages (n = 396) | Wald (X²) | p-value (AOR) | Percentages (n = 396) | Wald (X²) | p-value (AOR) |
| Sealing food items separately before chilling. | Never | 16.2 | 0.596 | 0.440 (0.276) | 15.9 | 0.081 | 0.775 (1.406) |
| | Sometimes | 57.6 | 0.528 | 0.467 (0.348) | 42.2 | 3.310 | 0.069 (4.159) |
| | Always | 26.3 | – | – | 41.9 | – | – |
| Chilling perishable food items without delay. | Never | 30.3 | 2.736 | 0.098 (0.074) | 19.7 | 1.254 | 0.263 (0.414) |
| | Sometimes | 48 | 3.779 | 0.052 (0.09) | 45.2 | 0.272 | 0.602 (0.705) |
| | Always | 21.7 | – | – | 35.1 | – | – |

The dummy variables indicated by hyphen (-) is a reference category in the omnibus statistical significance test

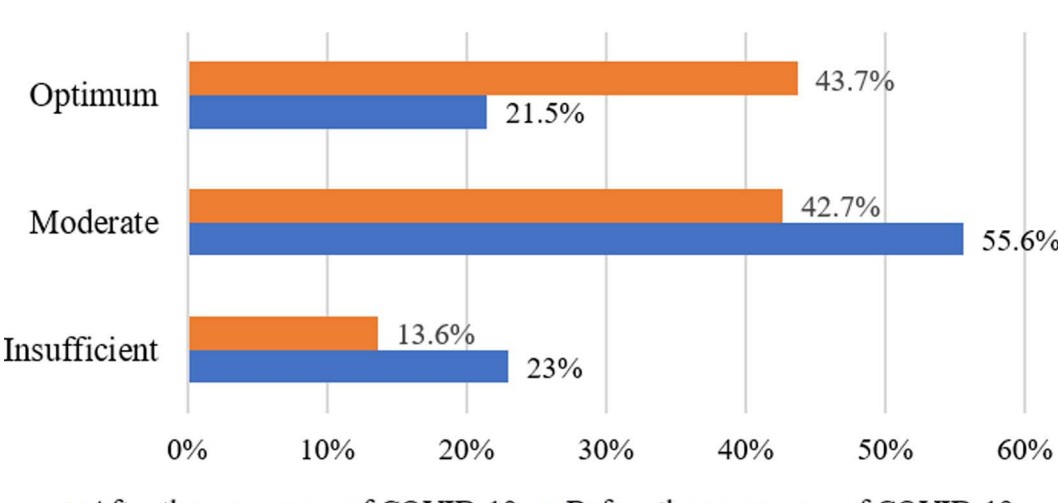

**Fig 2. Food safety practice of individuals before and after COVID-19 emergence.**

It is argued that since individuals' awareness levels were boosted during COVID-19 [33], there was no significant variation among individuals' food safety practices, and no statistically significant effect was observed in the present study. In the same way, the study conducted in Riyadh [8] reported that COVID-19 prevention practices indirectly benefited the food safety sector, and individuals' food safety awareness levels had statistically significant associations with their food safety practices.

In contrast to the previous situation, individuals' food safety practices after the emergence of COVID-19 were significantly linked to all dummy variables of educational status, with an adjusted odds ratio (AOR) of 1.291 for those with 'no formal education,' 'primary school,' and 'secondary school' education levels (Table 1), using 'higher education' qualification holders as the reference group. This suggests that participants with higher education qualifications had greater compliance with food safety practices compared to other educational categories. In other words, as educational levels increased, participants' adherence to recommended food safety standards also improved. The food safety practice differences

among educated and non-educated participants might be due to exposure to and understanding differences in food safety information. As found by previous studies [33,35,36], participants with formal education had higher food safety awareness levels than non-educated individuals. Unlike the current findings, a study conducted in Indonesia [37] found no statistically significant relationship between individuals' food safety practices and their educational level. This discrepancy is believed to stem from variations in participants' demographic factors, attitudes, and practices related to food safety [38], as well as differences in food handling and eating habits.

Before the emergence of COVID-19, all the dummy variables (no dependent, couple, have one child, and have 2–4 children) of participants' living arrangements concerning the dummy variable 'have more than 5 children at home' had statistically significant associations with the food safety practices of individuals. However, individuals' living arrangements did not influence their food safety practices after the emergence of COVID-19. The differences in food safety practices before and after the pandemic indicate that, prior to COVID-19, an increase in family size was associated with a greater violation of recommended food safety practices. However, following the outbreak, participants' compliance with these practices improved due to the infection prevention and control measures implemented for COVID-19. The assessment of pre-COVID-19 food safety practices confirmed that as family size increased, adherence to food safety guidelines decreased. Since COVID-19 affected individuals regardless of family size and extensive awareness campaigns were conducted through various platforms, including social media, community meetings, and village outreach programs [34,37,39], it is not surprising that no significant differences were observed in food safety practices post-COVID-19.

The type of work the participants used for income generation had a significant statistical association with their food safety practices pre-COVID-19. Except for non-governmental employees, all other dummy variables of the variable 'work for income generation' had a significant statistical association with the participants' food safety practices during the COVID-19 pandemic. Since the non-governmental organization employees had applied optimum food safety practices during the emergence of COVID-19, there was no variance observed in this work category (Table 1). Those people employed in governmental and non-governmental organizations had higher education qualifications and had optimum food safety practices as compared with other work categories, like casual and daily laborers. Comparable to the current finding, higher education degree students had better food safety knowledge and practice than lower-grade students in Jordan [36].

### Pre and post COVID-19 emergence comparative food safety practices

The individuals' hygienic practice in the present study was lower than those in Mettu and Bedelle towns [40] but higher than the findings reported by Abdi et al., [41]. These differences may be attributed to individuals' levels of food safety awareness, the availability of food safety standards across different geographic locations, and the enforcement capacity of food safety regulators. All the cleaning-related questions (predictive variables) had statistically significant effects on the food safety practices of individuals before the emergence of COVID-19. Since the AOR of all predictor variables of cleaning were less than one (Table 2), the increase in the frequency of food safety practices (never and sometimes dummy variables) reduced the optimum food safety practices of individuals. Similar to the pre-COVID-19 period, individuals' handwashing practices before, during, and after food preparation in the post-COVID-19 period remained significantly associated with their overall food safety practices. The same was observed for handwashing after using the toilet, with adjusted odds ratios (AOR) of 0.008 and 0.066 for "never" and "sometimes" responses, respectively. This indicates that individuals who reported "never" practicing or "sometimes" practicing handwashing had a 0.8% and 6.6% reduction, respectively, in compliance with recommended food safety practices. The dummy variable 'never' with reference to the response 'always' in the predictive variable of using soap during hand and utensil washing had a statistically significant association with the pooled food safety practices of individuals, both before and after the emergence of COVID-19. This finding was comparable to other studies [42,43] conducted in distinct locations within Ethiopia.

Before the emergence of COVID-19, more than half (57.3%) of individuals stored cooked and uncooked food at the recommended refrigerator temperature only "occasionally/sometimes." However, during the pandemic, this occasional practice dropped to 41.7%, while the percentage of individuals who "always" followed proper food storage guidelines increased from 26.3% to 42.4%. This suggests that COVID-19 had an indirect yet positive impact on individuals' food storage habits. Additionally, adherence to recommended refrigeration temperatures had a statistically significant effect on overall food safety practices during the pandemic (Table 2). Consistent with this finding, a study by Prasetya et al. [9] reported that individuals' food storage practices, which were previously inadequate, improved after the pandemic. These differences may be attributed to variations in participants' awareness levels, food safety governance systems, and consumer behavior and preferences [44].

The optimum food safety practices of individuals increased to 43.7% after the emergence of COVID-19 compared to pre-COVID-19 (21.5%). Similarly, to the present finding, the study conducted in Debre Markos City [42] reported that the food safety practices of individuals were optimum (54%) even before the emergence of COVID-19. The improvement in food safety practices might be due to the strict COVID-19 prevention measures applied during the pandemic [39]. The same research finding [39] also confirmed that the COVID-19 infection prevention and control strategies indirectly improved the food safety practices of individuals. The moderate (55.6%) and insufficient (23%) food safety practices before the emergence of COVID-19 were reduced to 42.7% and 13.6% post-COVID-19, respectively (Fig 2). In summary, the proportion of individuals with moderate (12.9%) and insufficient (9.4%) food safety practices before the emergence of COVID-19 shifted toward optimal food safety practices during the first three years of the pandemic. This improvement in individuals' previously inadequate food safety habits is supported by findings from other studies [29,30,33–35,39].

## Policy implications

While the COVID-19 pandemic caused crises across various sectors, including food security [45], health [46], and socio-economic dimensions [47], it also led to positive improvements in food safety. These improvements stemmed from the advocacy, enforcement, and implementation of COVID-19-related infection prevention and control measures [7,29,30,33,35,39,40]. As demonstrated by the current study and supported by other research [42,43], the recommended food safety steps outlined by the CDC [19] showed significant improvement in post-pandemic food safety assessments compared to pre-pandemic food safety practices.

Beyond the food safety sector, COVID-19 also had positive effects on physical, mental, social, and digital health due to the preventive and control measures implemented during the pandemic [6]. This highlights that COVID-19 had both negative and positive impacts on human well-being, emphasizing the need to sustain its positive outcomes while mitigating the crises it caused.

A policy intervention promoting compliance with post-pandemic food safety standards is now more crucial than ever. Policymakers should develop standardized and achievable food safety regulations, while food safety regulators must strive to maintain the unexpected improvements brought about by COVID-19. Additionally, food security regulators should leverage the infection prevention and control awareness generated during the pandemic to ensure long-term food safety improvements.

To achieve this, continuous community awareness campaigns, the development of food safety guidelines, and ongoing training on their implementation should be key priorities for food sustainability policies. Further research is also necessary to explore the sustainability of improved food safety practices. The contribution of research and academic institutions in investigating ways to maintain these improvements and in formulating and implementing effective food safety standards will be essential.

## Limitation of the study

This study excluded individuals who were not food buyers or not present in the open-air market during data collection. Consequently, it may have overlooked data from individuals responsible for food handling in other settings who may have different food safety practices, potentially leading to participant selection bias. Furthermore, food safety is a multifaceted issue influenced by various biological, socio-economic, and political factors, making it difficult to obtain a fully representative sample.

Since the data relied on individuals' retrospective recollection of their actions before and after COVID-19 was confirmed in Ethiopia (13 March 2020), there is a risk of recall bias. Additionally, as the study was based on self-reported information, there is a possibility of self-reporting bias, where participants might exaggerate, underreport, or selectively provide socially desirable responses. Moreover, because the survey was conducted in Amharic, only four cities with Amharic-speaking populations were included, excluding other regions of the country. As a result, food safety practices in other parts of Ethiopia may differ, and these findings may not be representative of the entire population.

## Conclusion

The assessment of individuals' food safety practices was conducted for both pre- and post-COVID-19 periods. The results indicate that post-pandemic food safety practices have improved compared to those before the pandemic. Participants showed significantly higher compliance with recommended food safety measures after the pandemic, suggesting that COVID-19 led to unintended yet beneficial changes in food safety practices. These positive outcomes should be preserved and promoted in the post-pandemic era, as they may play a vital role in preventing future public health crises.

The findings of this study are essential for national food safety and drug authorities in establishing and enforcing food safety standards. Strict adherence to food safety regulations can help curb the spread of endemic, epidemic, and pandemic diseases. Since the food value chain provides a conducive environment for pathogen formation, growth, and transmission, strengthening food safety measures can have far-reaching benefits for public health.

It is recommended that academic and research institutions advocate for food safety initiatives and educate both students and the broader community on infection prevention, food value chain management, traceability, food safety regulations, and enforcement strategies. Additionally, collaboration between public and private sectors is necessary to achieve shared food safety goals. Further research is needed to explore the long-term sustainability of the improvements in food safety practices observed after COVID-19. These studies could be guided by the theory of reasoned action, which focuses on transforming cautious food safety practices into lasting habits.

## Supporting information

**S1 File. Individuals' food safety practice assessment questions, variable codes, and descriptions, before and after the emergence of COVID-19.**
(XLSX)

## Author contributions

**Conceptualization:** Daniel Teshome Gebeyehu, Leah East, Stuart Wark, Md Shahidul Islam.

**Data curation:** Daniel Teshome Gebeyehu.

**Formal analysis:** Daniel Teshome Gebeyehu.

**Investigation:** Daniel Teshome Gebeyehu, Leah East, Stuart Wark, Md Shahidul Islam.

**Methodology:** Daniel Teshome Gebeyehu, Leah East, Stuart Wark, Md Shahidul Islam.

**Project administration:** Daniel Teshome Gebeyehu, Leah East, Stuart Wark, Md Shahidul Islam.

**Resources:** Daniel Teshome Gebeyehu, Leah East, Stuart Wark, Md Shahidul Islam.

**Software:** Daniel Teshome Gebeyehu.

**Supervision:** Leah East, Stuart Wark, Md Shahidul Islam.

**Validation:** Leah East, Stuart Wark, Md Shahidul Islam.

**Visualization:** Daniel Teshome Gebeyehu.

**Writing – original draft:** Daniel Teshome Gebeyehu.

**Writing – review & editing:** Daniel Teshome Gebeyehu, Leah East, Stuart Wark, Md Shahidul Islam.

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
