## [Decision Letter · Decision Letter 0]

7 Aug 2024

PONE-D-24-01799Food safety practices of individuals before and after the emergence of COVID-19: a pre and post comparative analysisPLOS ONE

Dear Dr. Gebeyehu,

Thank you for submitting your manuscript to PLOS ONE. After careful consideration, we feel that it has merit but does not fully meet PLOS ONE’s publication criteria as it currently stands. Therefore, we invite you to submit a revised version of the manuscript that addresses the points raised during the review process.

We look forward to receiving your revised manuscript.

Kind regards,

Werku Etafa

Academic Editor

PLOS ONE

Reviewers' comments:

Reviewer's Responses to Questions

**Comments to the Author**

1. Is the manuscript technically sound, and do the data support the conclusions?

Reviewer #1: Partly

2. Has the statistical analysis been performed appropriately and rigorously?

Reviewer #1: Yes

3. Have the authors made all data underlying the findings in their manuscript fully available?

Reviewer #1: Yes

4. Is the manuscript presented in an intelligible fashion and written in standard English?

Reviewer #1: Yes

5. Review Comments to the Author

Reviewer #1: 1. much volume of result narration at the abstract section, better to reduce

2. From January 2020 to December 2023 is only three years not four years

3. Better to include non-response rate in the methodology section

4. How the 10th food buyer in each open market is selected?

5. Operationally define what 'open market' is.

6. How can ' Language" be a barrier in study? Why not translate to different local languages?

7. There are sample size fallacies as 385 in the methodology part and 396 in the result section.

6. PLOS authors have the option to publish the peer review history of their article (what does this mean? ). If published, this will include your full peer review and any attached files.

**Do you want your identity to be public for this peer review?** For information about this choice, including consent withdrawal, please see our Privacy Policy .

Reviewer #1: **Yes: ** Yohannes Shumet Yimer

---

## [Author Response · Author response to Decision Letter 1]

11 Aug 2024

The manuscript is revised based on the reviewer's comments and point by point response is separately attached with a file name of 'Response to Reviewers'.

---

## [Decision Letter · Decision Letter 1]

5 Nov 2024

PONE-D-24-01799R1Food safety practices of individuals before and after the emergence of COVID-19: a pre and post comparative analysisPLOS ONE

Dear Dr. Gebeyehu,

Thank you for submitting your manuscript to PLOS ONE. After careful consideration, we feel that it has merit but does not fully meet PLOS ONE’s publication criteria as it currently stands. Therefore, we invite you to submit a revised version of the manuscript that addresses the points raised during the review process.

We look forward to receiving your revised manuscript.

Kind regards,

Adera Debella Kebede, MSC

Academic Editor

PLOS ONE

Reviewers' comments:

Reviewer's Responses to Questions

**Comments to the Author**

1. If the authors have adequately addressed your comments raised in a previous round of review and you feel that this manuscript is now acceptable for publication, you may indicate that here to bypass the “Comments to the Author” section, enter your conflict of interest statement in the “Confidential to Editor” section, and submit your "Accept" recommendation.

Reviewer #2: All comments have been addressed

2. Is the manuscript technically sound, and do the data support the conclusions?

Reviewer #2: Yes

3. Has the statistical analysis been performed appropriately and rigorously?

Reviewer #2: Yes

4. Have the authors made all data underlying the findings in their manuscript fully available?

Reviewer #2: Yes

5. Is the manuscript presented in an intelligible fashion and written in standard English?

Reviewer #2: Yes

6. Review Comments to the Author

Reviewer #2: Dear Authors,

Thank you for your valuable contribution to the understanding of food safety practices before and after the COVID-19 pandemic, particularly in the Ethiopian context. The manuscript is well-researched, and the findings are significant for public health interventions. Below is a summary of my feedback based on the review:

1. Technical Soundness & Data Validity:

Suggestions:

Clarify some of the more technical statistical terms for non-expert readers, such as "Wald X²" and "dummy variables." Brief explanations could help make the analysis more accessible to a wider audience.

2. Presentation and Language:

Simplify some of the longer sentences and correct minor punctuation errors to enhance clarity.

Ensure consistency in formatting numerical data and percentages.

3. Conclusion & Recommendations:

Consider expanding on how your findings can be integrated into national food safety policies or education programs. This could strengthen the practical implications of your research.

7. PLOS authors have the option to publish the peer review history of their article (what does this mean? ). If published, this will include your full peer review and any attached files.

**Do you want your identity to be public for this peer review?** For information about this choice, including consent withdrawal, please see our Privacy Policy .

Reviewer #2: **Yes: ** A Alyafei

---

## [Author Response · Author response to Decision Letter 2]

10 Nov 2024

I have already edited (location, age, and residential area variables are removed) and re-attached the supporting information file, as requested yesterday. Although I have edited it as requested, Paula Katrina A. Maderazo returned the manuscript to me again without mentioning any issues.

Please clarify what are you asking and what type of issue you found in our manuscript.

---

## [Decision Letter · Decision Letter 2]

20 Feb 2025

PONE-D-24-01799R2Food safety practices of individuals before and after the emergence of COVID-19: a pre and post comparative analysisPLOS ONE

Dear Dr. Gebeyehu,

Thank you for submitting your manuscript to PLOS ONE. After careful consideration, we feel that it has merit but does not fully meet PLOS ONE’s publication criteria as it currently stands. Therefore, we invite you to submit a revised version of the manuscript that addresses the points raised during the review process.

We look forward to receiving your revised manuscript.

Kind regards,

Adera Debella Kebede, MSC

Academic Editor

PLOS ONE

Additional Editor Comments:

Title: Food safety practices of individuals before and after the emergence of COVID-19: a pre- and post-comparative analysis

This study explores “Food safety practices of individuals before and after the emergence of COVID-19: a pre- and post-comparative analysis The authors have done a commendable job, and I congratulate them. I have some critical comments and some questions for the authors, which, if addressed, will increase the readability and understandability of the manuscript. I have put it section by section for ease of communication

Methods :

1. Please include the study period ( the same comments need to be incorporated in the section of methods also)

2. line 19: You have mentioned ….”three regional (Dessie, Kombolcha, and Debre Birhan) cities of Ethiopia” … it is crystal clear these are a pocket area in Amhara reginal state. Thus, you have state this as (Amhara Reginal State (Dessie, Kombolcha, and Debre Birhan)

3. line 21: you have stated that …”An equal number of participants were surveyed from metropolitan and regional cities”….why an equal number ? Have you assured the representativeness of the sample ?How you choose the food shops ?

4. Which software used for data entry and analysis should be indicated and when you declare statistical significance should be clearly mentioned

Result: change this to <Results >

Conclusion : should be <conclusions > and the conclusion should be based only on the current study findings, according to your study objective. Don’t put general ideas and common sense here. All the recommendations should align with your study findings.

Conceptual model :

1. Line 61: Fig 1: “Concept map of COVID-19’s impact on food safety and its indirect impact on health based on the findings of previous studies”……have a citation as you have taken from finding of previous studies

MATERIALS AND METHODS:

1 Generally. Reporting guidelines: Please ensure you follow the appropriate reporting guidelines when preparing your manuscript and submit the completed checklist as supplementary material. Please state in the methods which guidelines were consulted when preparing the manuscript. More information on reporting guidelines can be found on the EQUATOR website

2 Study design, sample size, and sampling procedure

3 The study seems that community type of study, if that is case why don’t use a design effect in calculating the sample size ?

4 If you said that , individuals were selected using random sampling methods ..How you create the sampling frame both for individuals and included open air market

5 Line 119-121: you have mentioned that …..”As the data collection was not self-administered and was conducted in face-to-face interviews by trained data collectors, ultimately there were no non-respondent participants in the survey. The researchers had expected a 3% non-response rate and therefore the total sample collected, including the expected nonresponse rate” ……I have afraid of non-response rate of 3% , you chose ? what if an individual’s withdraw in middle of data collection ? once agreed to fill and refuse to participate ? what if selected individuals absent from the works ? what if he/she overloaded with certain over job? How you tackled these factors ?

6 Please include both “study variables” and “operational definitions of terms “ as different sub titles

7 Ethical approval: Personally, I think the study conducted in Ethiopia and your target (study)populations are Ethiopian . But you have mentioned that you have obtained “Ethics approval from the University of New England Human Research Ethics Committee 133 (HREC) with an approval number: HE22-173” ….needs a further elaboration as you had to have an approval also from National ethics review Board of Ethiopia

Results

1 Please have a “Sub title “which clearly enhance the quality of your manuscript.

2 You have to report with Adjusted Odd Ratio

3 The results section needs tightening. For the descriptive section, either you report results as percentages or proportions (e.g., n/N). The way you have it now is confusing

Discussion

1 Generally, the discussion lacks depth, as it only compares the results across studies. When comparing the study findings with reports from the existing literature, it is recommended to comment on the reasons for the observed discrepancy based on the theoretical framework or give a plausible explanation. It is also required to indicate its clinical and theoretical implications on the topic of interest.

2 Further elaboration on statistical assumptions and policy implications would enhance transparency.

Strengths and Limitations of Study

1. Limitations should address potential biases (e.g., selection or recall bias), constraints in data collection (e.g., self-reporting or incomplete data), and generalizability of the findings to other populations or settings.

Conclusions and other section

1. The conclusion should be based only on the current study findings, according to your study objective. Don’t put general ideas and common sense here. All the recommendations should align with your study findings.

2. Finally, don’t forget attaching the most important section at end of your manuscript such as Data Sharing Statement, Ethical Approval, Acknowledgments, Authors’ Contributions, Funding Statement and Disclosure

Reviewers' comments:

Reviewer's Responses to Questions

**Comments to the Author**

1. If the authors have adequately addressed your comments raised in a previous round of review and you feel that this manuscript is now acceptable for publication, you may indicate that here to bypass the “Comments to the Author” section, enter your conflict of interest statement in the “Confidential to Editor” section, and submit your "Accept" recommendation.

Reviewer #2: All comments have been addressed

2. Is the manuscript technically sound, and do the data support the conclusions?

Reviewer #2: Yes

3. Has the statistical analysis been performed appropriately and rigorously?

Reviewer #2: Yes

4. Have the authors made all data underlying the findings in their manuscript fully available?

Reviewer #2: No

5. Is the manuscript presented in an intelligible fashion and written in standard English?

Reviewer #2: Yes

6. Review Comments to the Author

Reviewer #2: However, further elaboration on statistical assumptions and policy implications would enhance transparency. Data availability should be confirmed to ensure compliance with PLOS policies. Overall, the study is suitable for publication with minor refinements.

7. PLOS authors have the option to publish the peer review history of their article (what does this mean? ). If published, this will include your full peer review and any attached files.

**Do you want your identity to be public for this peer review?** For information about this choice, including consent withdrawal, please see our Privacy Policy .

Reviewer #2: **Yes: ** Anees Alyafei

---

## [Author Response · Author response to Decision Letter 3]

26 Feb 2025

There is no comment from reviewers. The comments from editor are addressed and "response to reviewer" file is attached separately.

---

## [Editor Report · Decision Letter 3]

12 Mar 2025

PONE-D-24-01799R3Food safety practices of individuals before and after the emergence of COVID-19: a pre- and post-comparative analysisPLOS ONE

Dear Dr.Teshome Gebeyehu ,

Thank you for submitting your manuscript to PLOS ONE. After careful consideration, we feel that it has merit but does not fully meet PLOS ONE’s publication criteria as it currently stands. Therefore, we invite you to submit a revised version of the manuscript that addresses the points raised during the review process.

We look forward to receiving your revised manuscript.

Kind regards,

Adera Debella Kebede, MSC

Academic Editor

PLOS ONE

Journal Requirements:

Additional Editor Comments:

Title: Food safety practices of individuals before and after the emergence of COVID-19: a pre- and post-comparative analysis

Thank you for addressing the major issues, here are also some points needs to consider and incorporated before deciding on the publication of your paper

Methods :

1. Inconsistency with your study period (line 20: 16 April 2023, and 31 June 2023; line 122: 6 April 2023 to 30 June 2023) please check it

2. Line 27 “The data was entered IBM SPSS version 28 and analyzed using ordinal logistic regression” ….how that ? if you entered the data using SPSS, how you control logical skipping and others . Why don’t you used EPI data software for data entry ? and moreover, once you amended the stated statement, move it to methods section of abstract .

MATERIALS AND METHODS:

1 Explicitly, you have employed STROBE Statement—Checklist of items as per comments but some of the points needs to be addressed in the details and you need to have a subtitle such for instance “Bias”…. Like recall bias, social desirability bias and how you mitigate it

2 Line 147-158: what you mentioned under the subtitle of “ study variables” need to moved to section of “operational definition “. Rather here you are expected your dependent (outcome variables) and independent variables

3 Line 160-188:Please re write section as “operational definitions of terms and measurement” . First define the listed term such as cleaning, separating and cooking, chilling etc. …. Then tell us how you measure it

---

## [Author Response · Author response to Decision Letter 4]

14 Mar 2025

A point by point response is attached.

---

## [Editor Report · Decision Letter 4]

18 Mar 2025

Food safety practices of individuals before and after the emergence of COVID-19: a pre- and post-comparative analysis

PONE-D-24-01799R4

Dear Dr.Teshome Gebeyehu,

We’re pleased to inform you that your manuscript has been judged scientifically suitable for publication and will be formally accepted for publication once it meets all outstanding technical requirements.

Kind regards,

Adera Debella Kebede, MSC

Academic Editor

PLOS ONE
---

## [Editor Report · Acceptance letter]

PONE-D-24-01799R4

PLOS ONE

Dear Dr. Gebeyehu,

I'm pleased to inform you that your manuscript has been deemed suitable for publication in PLOS ONE. Congratulations! Your manuscript is now being handed over to our production team.

Kind regards,

on behalf of

Dr. Adera Debella Kebede

Academic Editor

PLOS ONE